# Anti-Neuroinflammatory Effects of *Ginkgo biloba* Extract EGb 761 in LPS-Activated BV2 Microglial Cells

**DOI:** 10.3390/ijms25158108

**Published:** 2024-07-25

**Authors:** Lu Sun, Matthias Apweiler, Ashwini Tirkey, Dominik Klett, Claus Normann, Gunnar P. H. Dietz, Martin D. Lehner, Bernd L. Fiebich

**Affiliations:** 1Neuroimmunology and Neurochemistry Research Group, Department of Psychiatry and Psychotherapy, Medical Center-University of Freiburg, Faculty of Medicine, University of Freiburg, 79104 Freiburg, Germany; lusun.jane@gmail.com (L.S.); matthias.apweiler@uniklinik-freiburg.de (M.A.); ashwini.tirkey97@gmail.com (A.T.); dominik.klett@uniklinik-freiburg.de (D.K.); 2Department of Psychiatry and Psychotherapy, Medical Center-University of Freiburg, Faculty of Medicine, University of Freiburg, 79104 Freiburg, Germany; claus.normann@uniklinik-freiburg.de; 3Universitätsmedizin Göttingen, Robert-Koch-Str. 40, 37075 Göttingen, Germany; gdietz@gwdg.de; 4Dr. Willmar Schwabe GmbH & Co. KG, Willmar-Schwabe-Straße 4, 76227 Karlsruhe, Germany; martin.lehner@schwabe.de

**Keywords:** EGb 761, neuroinflammation, cytokines, chemokines, microglia, MAPK pathway, NF-κB pathway

## Abstract

Inflammatory processes in the brain can exert important neuroprotective functions. However, in neurological and psychiatric disorders, it is often detrimental due to chronic microglial over-activation and the dysregulation of cytokines and chemokines. Growing evidence indicates the emerging yet prominent pathophysiological role of neuroinflammation in the development and progression of these disorders. Despite recent advances, there is still a pressing need for effective therapies, and targeting neuroinflammation is a promising approach. Therefore, in this study, we investigated the anti-neuroinflammatory potential of a marketed and quantified proprietary herbal extract of *Ginkgo biloba* leaves called EGb 761 (10–500 µg/mL) in BV2 microglial cells stimulated by LPS (10 ng/mL). Our results demonstrate significant inhibition of LPS-induced expression and release of cytokines tumor necrosis factor-α (TNF-α) and Interleukin 6 (IL-6) and chemokines C-X-C motif chemokine ligand 2 (CXCL2), CXCL10, c-c motif chemokine ligand 2 (CCL2) and CCL3 in BV2 microglial cells. The observed effects are possibly mediated by the mitogen-activated protein kinases (MAPK), p38 MAPK and ERK1/2, as well as the protein kinase C (PKC) and the nuclear factor (NF)-κB signaling cascades. The findings of this in vitro study highlight the anti-inflammatory properties of EGb 761 and its therapeutic potential, making it an emerging candidate for the treatment of neuroinflammatory diseases and warranting further research in pre-clinical and clinical settings.

## 1. Introduction

Over the decades, neuroinflammation has been identified as a prevalent phenomenon in several brain-related pathologies [1,2,3]. Extensive research on the role of neuroinflammation is conducted to understand the molecular mechanisms involved in neuropsychiatric disorder such as Alzheimer’s disease (AD), Parkinson’s disease (PD) and depression [4]. A large body of evidence suggests that neuronal degeneration, as observed in diseases, such as AD and PD, is significantly driven and influenced by neuroinflammation [5,6]. These inflammatory responses are orchestrated by microglia, the central nervous system (CNS)’s resident immune cells, that are integral to immune surveillance and responses, phagocytosis, antigen presentation and maintaining brain homeostasis [7]. Microglial cells are highly plastic in nature due to their ability to assume diverse phenotypes and their dualistic roles in neuronal injury and recovery [8]. Certain factors, such as aggregated protein deposits, pro-inflammatory cytokines, free radicals, neurotoxins, etc., can trigger the conversion of resting, ramified microglia to their activated amoeboid form [9,10]. Activated microglia release pro-inflammatory mediators, especially cytokines and chemokines, which further enhance neuroinflammation [11]. Therefore, targeting neuroinflammation in neuropsychiatric disorders may provide a beneficial strategy for therapy.

Herbal medications derived from *Ginkgo biloba* leaf extracts are one of the most popular and best studied herbal medicines that are used clinically in the treatment of a number of different indications. *Ginkgo biloba* extracts are used as an accepted treatment option for cardiovascular and cerebrovascular indications in different countries [12] as well as for ameliorating cognitive impairment and quality of life in dementia sufferers [13]. EGb 761 is a special quantified leaf extract of *Ginkgo biloba* [14] that has been most extensively used in clinical trials, especially with focus on dementia and cognitive impairment. Several modern clinical studies have shown that EGb 761 exerts therapeutic benefits in mild neurocognitive impairment [15], dementia [16], dementia with cerebrovascular disease [17] and dementia with neuropsychiatric disorders [18]. Mechanistically, a number of different pharmacological activities with potential relevance for the clinical use have been reported from clinical and non-clinical studies, including vasoregulatory activity, improved blood flow, anti-oxidative activity, reduction in amyloid oligomer and plaque formation and toxicity, improvement of neuroplasticity, increase in neurotransmitter levels and protection against mitochondrial dysfunction [19,20,21,22,23,24,25].

Additionally, we previously showed that EGb 761 demonstrated anti-neuroinflammatory activity in LPS-activated primary microglia cells. EGb 761 reduced neuroinflammation by targeting the COX-2/PGE_2_ pathway. It was suggested that the observed anti-neuroinflammatory effects could potentially explain the established clinical cognitive efficacy in AD, vascular and mixed dementia. However, other important inflammatory parameters such as cytokines and chemokines were not investigated and are therefore the focus of the current study. Here, we investigated effects of EGb 761 on the expression and release of the chemokines CXCL2, CXCL10, CCL2, CCL3 and the cytokines IL-6 and TNFα in LPS-stimulated BV2 microglial cells. We further examined the possible intracellular signaling pathways involved in the effects of EGb 761.

## 2. Results

### 2.1. Effects of EGb 761 on Cell Viability in BV2 Microglial Cells

The cell viability of BV2 microglial cells after treatment with EGb 761 was examined using an MTT assay. As presented in Figure 1, EGb 761 did not show cytotoxic effects in LPS-stimulated BV2 microglia cells, as evidenced by the conversion of MTT to formazan, compared to untreated cells. Conversely, the positive control of 20% Ethanol significantly induced cell death. Since EGb 761 did not affect cell viability and thus showed no toxic effect, we proceeded with doses up to 500 µg/mL for further experiments.

### 2.2. Effects of EGb 761 on LPS-Induced TNF-α Expression and Release in BV2 Microglial Cells

The neuroinflammatory role of TNF-α, as a key pro-inflammatory cytokine in the CNS [26], has been demonstrated in multiple studies. We assessed the effects of EGb 761 on TNF-α expression and release in LPS-stimulated BV2 microglial cells. BV2 microglial cells were pre-treated with different concentrations of EGb 761 (10 to 500 µg/mL) for 30 min before being stimulated with LPS (10 ng/mL) for 4 h (mRNA expression) or 24 h (protein release). The expression and release of TNF-α was strongly induced by LPS (Figure 2) compared to untreated cells. EGb 761 exhibited a concentration-dependent decrease in TNFα levels induced by LPS, achieving significant reduction at 500 µg/mL (Figure 2A). However, basal levels of TNF-α expression were not attained. In contrast, EGb 761 significantly inhibited LPS-induced TNF-α release in all doses tested (Figure 2B). The discrepancy between the effects on mRNA and protein levels might be due to the time point chosen for the RNA analysis or a post-transcriptional effect.

### 2.3. Effects of EGb 761 on LPS-Induced IL-6 Expression and Release in BV2 Microglial Cells

Since a large body of research reports associations between neurodegeneration and IL-6 [27,28], the effects of EGb 761 on IL-6 expression and secretion in BV2 microglial cells following LPS stimulation were determined. BV2 cells were pre-incubated with different concentrations of EGb 761 (10 to 500 µg/mL) for 30 min. Afterwards LPS was added for 4 h (expression) or 24 h (release). As shown in Figure 3, LPS significantly induced IL-6 expression and release compared to untreated cells. Pre-treatment with EGb 761 revealed a significant albeit partial inhibition on LPS-induced of IL-6 expression (Figure 3A) and release (Figure 3B) starting at concentrations of 250 µg/mL.

### 2.4. Effects of EGb 761 on LPS-Induced CCL2 Expression and Release in BV2 Microglial Cells

In the CNS, chemokines play a vital role in the pathology of neurodegenerative disorders. We therefore investigated the effects of EGb 761 on several chemokines induced by LPS in BV2 microglial cells.

First, we investigated the impact of EGb 761 on LPS-induced expression and synthesis of the chemokine (C-C motif) ligand 2 (CCL2), also known as monocyte chemoattractant protein 1 (MCP1). Treatment of BV2 cells with LPS (10 ng/mL) for 4 h (expression) or 24 h (release) caused a significant induction of CCL2 expression and release (Figure 4). Pre-treatment with EGb 761 strongly and significantly inhibited LPS-induced CCL2 expression and release in all concentrations tested (Figure 4). An amount of 500 µg/mL EGb 761 reduced CCL2 release even beneath basal levels.

### 2.5. Effects of EGb 761 on LPS-Induced CCL3 Expression and Release in BV2 Microglial Cells

Next, we focused on the effects of EGb 761 on LPS-induced chemokine (C-C motif) ligand 3 (CCL3), also referred to as macrophage inflammatory protein 1α (MIP-1α). CCL3, a key chemokine in neuroinflammation [29,30], was significantly expressed and synthesized in response to LPS in BV2 microglial cells (Figure 5). EGb 761 showed a slight inhibition of LPS-induced expression of CCL3, with significant effects at concentrations of 100 and 500 µg/mL (Figure 5A). In contrast, LPS-induced CCL3 release was potently and concentration-dependently prevented by EGb 761, even below basal levels (Figure 5B). The quantitative differences between the effects on mRNA and protein levels might be, as already discussed for TNFα, due to the different time points chosen for the RNA vs. protein analysis or a post-transcriptional effect.

### 2.6. Effects of EGb 761 on LPS-Induced CXCL2 Expression and Release in BV2 Microglial Cells

CXCL2, also known as macrophage inflammatory protein-2α (MIP-2α), growth oncogene-2 (Gro-2) or growth-regulated protein beta (GRO-β), is a small chemokine involved in inflammation and is expressed by various cell types, including microglia, astrocytes and neurons [31]. LPS potently induced CXCL2 expression and synthesis in BV2 microglial cells (Figure 6). EGb 761 showed a weak concentration-dependent inhibition of LPS-induced CXCL2 expression, reaching statistical significance at 500 µg/mL (Figure 6A) and also produced a significant, concentration-dependent reduction in LPS-induced CXCL2 synthesis in BV2 microglia cells (Figure 6B).

### 2.7. Effects of EGb 761 on LPS-Induced CXCL10 Expression and Release in BV2 Microglial Cells

CXCL10, also named as interferon-inducible protein (IP-10), is involved in chemotaxis and the modulation of inflammatory responses in immune cells and is suggested to have a role in neurodegeneration [32]. LPS substantially upregulated CXCL10 expression and synthesis in BV2 microglial cells (Figure 7). EGb 761 showed a weak but concentration-dependent inhibition of LPS-induced CXCL10 expression reaching statistical significance starting at 100 µg/mL (Figure 7A). EGb 761 exerted a potent and significant inhibition of LPS-induced CXCL10 release with a maximal inhibition of 50% using a concentration of 500 µg/mL (Figure 7B).

### 2.8. Effects of EGb 761 on Phosphorylation of PKCβ, p38 MAPK, ERK 1/2, and NF-κB

The MAPK signaling pathway, consisting of SAPK/JNK, p38 MAPK and ERK1/2, is integral to the regulation of various cellular activities. Increasing evidence highlights the crucial involvement of MAPK signaling in neuroinflammation and neurodegeneration [33]. This study explored the effects of EGb 761 on LPS-induced phosphorylation and thus activation of ERK1/2 and p38 MAPK in BV2 microglial cells.

LPS stimulation induced the phosphorylation of ERK1/2 and p38 MAPK (Figure 8). The pre-incubation of EGb 761 for 30 min, concentration-dependently decreased LPS-induced phosphorylation of ERK1/2 and p38 MAPK down to basal phosphorylation levels, with significant effects using the highest dose of 500 µg/mL for ERK1/2 (Figure 8A) and the two highest concentrations of 250 µg/mL and 500 µg/mL for p38 MAPK (Figure 8B).

PKC, a family of phospholipid-dependent serine/threonine kinases, is classed into three subfamilies based on structural and activatory properties. We explored the effects of EGb 761 on the PKC βII Ser660 phosphorylation site. The antibody used also detects selective phosphorylation of homologous sites on *PKC α, β I, β II, δ, ε, η* and *θ isoforms*. Figure 9A showed that LPS stimulation increased the levels of phospho-PKC (pan) (βII Ser660) and EGb 761 concentration-dependently decreased phosphorylation of PKC (pan) (βII Ser660), achieving statistical significance at 500 µg/mL (Figure 9A).

NF-κB, a family of dimeric transcription factors, plays a crucial role in orchestrating inflammatory responses [34]. Therefore, we investigated the effects of EGb 761 on the phosphorylation of NF-κB subunit p65. LPS induced the phosphorylation of NF-κBp65 and EGb 761 exhibited a concentration-dependent inhibition of the p65 NF-κB phosphorylation and exerted significant effects at the maximum concentration of 500 µg/mL (Figure 9B).

## 3. Discussion

The current study demonstrates anti-neuroinflammatory properties of the widely used *Ginkgo biloba* extract EGb 761 in LPS-stimulated BV2 microglial cells. EGb 761 exerts notable inhibition of LPS-induced gene expression and protein synthesis of the cytokines IL-6 and TNF-α, along with chemokines including CCL2, CCL3, CXCL2 and CXCL10. In addition, our findings suggest that these anti-inflammatory properties might be mediated via PKC/MAPK and NF-kB signaling. Overall, EGb 761 demonstrates considerable potential for mitigating inflammatory responses in BV2 microglial cells.

As the resident immune cells of the CNS, microglia are integral players in neuroinflammation and associated neurodegenerative diseases [35]. Microglia are activated by external or internal pathogens, such as pathogen-associated molecular patterns (PAMPs), damage-associated molecular patterns (DAMPs), protein aggregates or oxidative stress. This microglial activation leads to the release of inflammatory mediators, such as cytokines and chemokines, which contribute to the neuroinflammatory response [36]. While acute neuroinflammation aims to clear pathogens and promote tissue repair, chronic and dysregulated microglial activation exacerbates neuronal damage and ultimately contributes to neurodegenerative processes. In neurodegenerative diseases such as AD, amyotrophic lateral sclerosis (ALS) and PD, sustained neuroinflammation perpetuated by activated microglia is a hallmark feature [37,38]. Moreover, in cognitive post-COVID-19 symptoms, inflammatory mechanisms are also likely to play a substantial role [39,40] Activated microglia not only exacerbate neurodegeneration by inducing neuronal death but also impair neurogenesis and synaptic function [41]. Understanding the dualistic role of microglia in neuroprotection and neurotoxicity is crucial for the development of highly specific pharmacotherapies aiming to slow or halt the progression of neurodegenerative diseases by targeting neuroinflammation.

The specialized extract EGb 761 from *Ginkgo biloba* leaf has demonstrated efficacy in improving cognitive function and neuropsychiatric symptoms in dementia patients [15,16,17,18], and it has also been suggested to be beneficial in cognitive post-acute COVID-19 syndromes [42]. The positive impact of EGb 761 on the CNS are attributed to its diverse constituents, acting synergistically on various targets [43] and clinical efficacy has not been attributed to a defined single mechanism of action. Numerous pharmacological activities have been shown mostly from non-clinical models such as anti-oxidative activity, regulation of blood flow, neuroprotection, improvement of neurogenesis and protection against mitochondrial dysfunction [19,20,21,22,23,24,25]. Our present data on the inhibitory activity of the *Ginkgo biloba* leaf extract EGb 761 on LPS-induced microglial cytokine and chemokine production suggest further advantageous effects of EGb 761 on neuroinflammation as a potential pathomechanism contributing to the loss of neuronal function in dementia. These data confirm and extend results from a previous study where EGb 761 exerted anti-inflammatory activities in primary rat microglial cells by influencing the COX/PGE_2_ pathway, significantly attenuating the LPS-induced production of pro-inflammatory mediators, such as PGE_2_, TNF-α, IL-6 and IL-1ß [26]. Substantial evidence suggests a multi-functional activity of EGb 761, including anti-inflammatory activities, which may be attributed to the integrated actions of its terpene lactones (ginkgolides and bilobalide) and flavonoid constituents [44]. It has been reported that ginkgolides exert inhibitory effects on the production of the pro-inflammatory cytokines TNF-α and IL-1 in primary rat microglial cultures stimulated with LPS [45]. Furthermore, EGb 761 decreases iNOS expression partially mediated via the suppression of p38 MAPK, which is essential for iNOS expression [46]. This property of EGb 761 is particularly noteworthy, especially considering the role of NO in neuroinflammation. In the current study, we confirmed the inhibition of IL-6 and TNF-α expression and synthesis after EGb 761 treatment in LPS-stimulated BV2 microglial cells. Additionally, the expression and synthesis of different chemokines (CXCL2, CXCL10, CCL2 and CCL3) were affected by EGb 761. Chemokines and their receptors, playing a critical role in the regulation of inflammatory responses by orchestrating the migration and activation of immune cells to sites of tissue damage or infection, have emerged as pivotal targets for immuno-interventions in inflammation [46,47]. For example, CXCL10 is a key chemokine triggering inflammatory molecular processes and exerting pro-inflammatory effects by recruiting pro-inflammatory cells and pathways [48]. Concerning the analysis of Ginkgo effects on intracellular signaling pathways, the inhibition of LPS-activated p38 MAPK and Akt signaling has been reported in venous endothelial cells (VEC) [49]. In our present study, we investigated pathways possibly involved in the synthesis of cytokines and chemokines in BV-2 mouse microglial cells to elucidate the molecular mechanisms of the effects of EGb 761 on inflammatory processes pertinent to neuroinflammation.

Based on a previous study, LPS induces pro-inflammatory responses via PKC, MAPK and NF-kB signaling in BV2 cells [50]. Here, we validated the involvement of PKC/MAPK and NF-kB signaling in these inflammatory processes. These signaling cascades are pivotal for regulating cell proliferation, immune responses and the synthesis of inflammatory mediators [51,52,53]. Several studies suggest that PKC, a family composed of phospholipid-dependent serine/threonine kinases, serves as a key regulator of inflammatory responses [54,55]. PKC isoforms (PKCs), integral to various signal transduction pathways, are implicated in regulating a multitude of cellular functions [56,57]. While there are at least 11 isoforms of PKC known, this study specifically measures the levels of endogenous PKC α, βI, βII, δ, ε, η and θ isoforms when phosphorylated at a carboxy-terminal residue homologous to serine 660 of PKC β II. It is reported that PKC ζ plays a crucial role in both activating NF-kB by regulating RelA and exerting anti-inflammatory effects by modulating the IL-4 signaling pathway [54]. Therefore, further investigations about effects of EGb 761 on different PKC isoforms and phosphorylation of downstream signaling molecules are necessary. Additionally, EGb 761 has been shown to be an inhibitor of p38 MAPK and NF-kB activity in LPS-stimulated RAW 264.7 macrophages [46]. However, in our previous study, we found that primary rat microglial cells stimulated with LPS exerted a significant degradation of IκBα, a pivotal event at the beginning of the NF-kB signaling pathway. Treatment with EGb 761 did not substantially reverse LPS-induced IκBα degradation, thereby failing to prevent phosphorylation and subsequent nuclear translocation of the NF-κBp65 subunit, although there was a slight trend towards reduced IκBα degradation, this effect did not reach statistical significance. Furthermore, the activation of three MAPKs (SAPK/JNK, p38 and p42/44 MAPK) was not inhibited by EGb 761 [26]. In the current study, we showed that the anti-inflammatory effects of EGb 761 appear to rely on the modulation of PKC/MAPK and NF-κB signaling. The variability between the reported experimental results might be caused by the utilization of diverse cell types and varying microenvironmental conditions. One of the limitations of in vitro studies on CNS drug effects is the lack of assessing blood–brain barrier effects. However, for Ginkgo constituents such as ginkgolides A and B, bilobalide and Ginkgo flavones, relevant tissue levels in brain and effects on CNS neurotransmitter levels have been reported from in vivo models [24,58,59,60]. In addition, oral administration of EGb 761 significantly attenuated the depressive-like behavior and induction of hippocampal cytokines in LPS-injected mice, confirming the relevance of our in vitro study results with BV-2 microglia [61].

To summarize, we demonstrated that EGb 761 effectively suppresses the LPS-induced expression and secretion of the chemokines (CCL2, CCL3, CXCL2 and CXCL10) as well as cytokines (IL-6 and TNFα) in BV2 microglial cells. We identified p38 MAPK, ERK1/2, NF-κB and PKC as intracellular targets.

## 4. Materials and Methods

### 4.1. Herbal Extract EGb 761^®^

EGb 761^®^ (provided by Dr. Willmar Schwabe GmbH & Co. KG, Karlsruhe, Germany) is a dry extract from *Ginkgo biloba* leaves (drug to extract ratio 35-67:1, extraction solvent: acetone 60% (*w*/*w*)).

The extract was adjusted to 22–27% Ginkgo flavonoids, which was calculated as Ginkgo flavone glycosides and 5.4–6.6% terpene lactones consisting of 2.8–3.4% ginkgolides A, B, C and 2.6–3.2% bilobalide, and contained 4.5–9.5% proanthocyanidins and less than 5 ppm ginkgolic acids.

### 4.2. Chemicals

For this study, EGb 761 was dissolved in distilled water to give a stock concentration of 10 mg/mL. Further dilutions were also made with distilled water to obtain the final working concentrations of 10, 100, 250 and 500 μg/mL. Lipopolysaccharide (LPS) from *E. coli* (O127:B8; Sigma-Aldrich GmbH, Taufkirchen, Germany) was used in the final concentration of 10 ng/mL after dissolving it in distilled water.

### 4.3. BV2 Cell Culture 

BV2 microglial cells were kindly provided by Prof. Langmann (Department of Ophthalmology, University of Cologne, Cologne, Germany) and cultured in 1x RPMI 1640 medium containing 10% fetal calf serum (FCS; Bio and SELL GmbH, Feucht/Nürnberg, Germany), 2 mM L-glutamine and 1% penicillin/streptomycin (all cell culture solutions obtained by Gibco, Thermo Fisher Scientific, Bonn, Germany) at 5% CO_2_, 37 °C, in a humidified culture atmosphere. When approx. 90% confluency was attained, the cells were passaged with trypsin and reseeded to 6-, 12-, 24-, or 96-well plates or new cell culture flasks, respectively. On the next day, the medium was changed and after 1 h, the cells were stimulated for respective experiments.

### 4.4. MTT Cell Viability Assay

Cell viability was measured by quantitative colorimetric assay with MTT. BV2 microglial cells were seeded on 96-well cell culture plates and incubated for 20 h in 5% CO_2_ at 37 °C after treatment with different concentrations of EGb 761 (10–500 µg/mL) with or without LPS (10 ng/mL). After treatment, cells were incubated with 20 µL of MTT solution for 4 h. In the next step, the supernatant was removed and the resulting formazan crystals were solubilized in dimethylsulfoxide (DMSO). The absorbance of each well was measured at 595 nm by using an MRX^e^ micro-plate reader (Dynex Technologies, Denkerdorf, Germany).

### 4.5. Determination of Chemokine and Cytokine Production

BV2 microglial cells were pre-incubated with various concentrations of EGb 761 for 30 min. Thereafter, cells were treated with or without LPS (10 ng/mL) at the indicated concentrations for the next 24 h. Next, supernatants were collected and centrifuged at 1000× *g* for 5 min at 4 °C. Concentrations of CXCL2, CXCL10, CCL2, CCL3, IL-6 and TNFα were assessed in the supernatants with commercially available immunoassays (ELISA) kits (cat. no.: DY466, DY452, DY479, DY450, DY406, DY410; BioTechne, Wiesbaden, Germany), according to manufacturer’s instructions. The 96-well microplates were coated overnight with respective capture antibodies, blocked, washed and incubated with respective standards and supernatant dilutions, as pre-determined using several dilutions of the supernatants. Subsequently, detection antibody, Streptavidin-HRP, substrate solution and stop solution were added to each well in later steps. The optical density was measured at 450 nm using the MRX^e^ micro-plate reader.

### 4.6. RNA Isolation and Quantitative PCR

Expression of cytokines and chemokines in LPS-stimulated BV2 microglial cells was assessed by quantitative real-time PCR (qPCR) by determining the relative fold change in gene expression using the delta-delta Ct method. Cultured cells were pre-treated with the respective concentrations of EGb 761 for 30 min, followed by stimulation with LPS (10 ng/mL) for 4 h. RNA isolation was performed using the GeneMATRIX Universal RNA Purification Kit (cat. no.: E3598, Roboklon GmbH, Berlin, Deutschland), according to the manufacturer’s protocol. Then, cDNA was generated from 500 ng of total RNA in a 30 μL total reaction volume with initial denaturation at 70 °C (10 min; with random primers, biomers.net GmbH, Ulm, Germany) followed by an amplification cycle after addition of master mix (cDNA master mix: M-MLV reverse transcriptase (cat. no.: M1708), RNAsin^®^ Ribonuclease Inhibitor (cat. no.: N-251B), RT-Buffer M-MLV 5x (cat. no.: M531A); Promega GmbH, Mannheim, Germany). The synthesized cDNA was used as template for the real-time qPCR amplification carried out by the CFX384 real-time PCR detection system (Bio-Rad Laboratories GmbH, Feldkirchen, Germany) using SYBR green as fluorescent dye (Applied Biosystems™ SYBR^®^ Select Master Mix for CFX, cat. no.: 4472942; Thermo Fisher Scientific, Bonn, Germany). Glyceraldehyde 3-phosphate dehydrogenase (GAPDH) served as an internal control for sample normalization. The primer sequences are as follows:

GAPDH: Forward (Fwd): 5′-TGGGAAGCTGGTCATCAAC-3′/Reverse (Rev): 5′-GCATCACCCCATTTGATGTT-3′, CXCL2: Fwd: 5′-CCCTCAACGGAAGAACCAAAG-3′/Rev: 5′-GAGGCACATCAGGTACGATCCA-3′, CXCL10: Fwd: 5′-CAGTGGATGGCTAGTCCTAATTG-3′/Rev: 5′-ACTCAGACCAGCCCTTAAAGAAT-3′, CCL2: Fwd: 5′-TGATCCCAATGAGTAGGCTGG-3′/Rev: 5′-ACCTCTCTCTTGAGCTTGGTG-3′, CCL3: Fwd: 5′-TATTTTGAAACCAGCAGCCTTT-3′/Rev: 5′-ATTCTTGGACCCAGGTCTCTTT-3′, IL-6: Fwd: 5′-AGTTGCCTTCTTGGGACTGA-3′/Rev: 5′-TTCTGCAAGTGCATCATCGT-3′, and TNF-α: Fwd: 5′-CCCACGTCGTAGCAAACCACCA-3′/Rev: 5′-CCATTGGCCAGGAGGGCGTTG-3′. Specific primers were designed using Primer-BLAST and obtained by biomers.net GmbH (Ulm, Germany).

### 4.7. Western Blot

BV2 microglial cells were treated with EGb 761 for 30 min, then LPS (10 ng/mL) was added for 30 min. Next, cells were washed with cold phosphate-buffered saline (PBS) and lysed in lysis buffer (42 mM Tris-HCl, 1.3% sodium dodecyl sulfate, 6.5% glycerin, 100 μM sodium orthovanadate and 2% phosphatase and protease inhibitors). Protein concentration of the samples was measured using the bicinchoninic acid (BCA) protein assay kit (Thermo Fisher Scientific, Bonn, Germany) according to the manufacturer’s instructions. For Western blotting, 10 µg of total protein from each sample was subjected to 10% sodium dodecyl sulfate-polyacrylamide gel electrophoresis (SDS-PAGE) under reducing conditions. Afterwards, proteins were transferred onto polyvinylidene fluoride (PVDF) membranes (Merck KGaA, Darmstadt, Germany) with a pore size of 0.45 μm. After blocking with 1x ROTI^®^ Block (Carl Roth GmbH + Co. KG, Karlsruhe, Germany), membranes were washed in Tris-buffered saline (TBS) containing 0.1% Tween 20 (TBS-T). Primary antibodies used were phospho-p38 MAPK (1:1000; cat. no.: 9211S, Cell Signaling Technology, Danvers, MA, USA), phospho-p44/42 MAPK (1:1000; cat. no.: 9101S, Cell Signaling Technology), phospho-PKC (pan) (βII Ser660) (1:1000; cat. no.: 9371S, Cell Signaling Technology), phospho-NF-κB p65 (1:1000; cat. no.: 3033S, Cell Signaling Technology), and mouse anti-vinculin (1:20,000, Merck, Darmstadt, Germany). Primary antibodies were diluted in TBS-T with 5% of bovine serum albumin (BSA). Membranes were incubated with the respective primary antibody overnight at 4 °C followed by incubation with secondary antibodies. After extensive washing (three times for 15 min each in TBS-T), proteins were detected with either horseradish peroxidase (HRP)-coupled anti-mouse IgG (cat. no.: 401253, Merck KGaA, Darmstadt, Germany) or anti-rabbit IgG (cat. no.: 5450-0010, SeraCare, Milford, MA, USA) using enhanced chemiluminescence (ECL) reagents (GE Healthcare, Freiburg, Germany). Densitometry analysis was performed using ImageJ software (V1:48t, NIH, Stapleton, NY, USA), and vinculin control was used to confirm equal sample loading and normalization of the data.

### 4.8. Statistical Analysis

Statistical analyses were performed using Prism 5 software (GraphPad Software Inc., San Diego, CA, USA). Results are presented as percentage of the positive control (stimulation with LPS 10 ng/mL or untreated cells for MTT-assay). Values of all experiments are represented as mean ± SD (standard deviation) of at least three independent experiments and compared using one-way ANOVA with Dunnett’s post hoc test. The level of significance was set at * *p* < 0.05, ** *p* < 0.01, *** *p* < 0.001 with respect to the positive control.

## 5. Conclusions

In conclusion, our study demonstrated that EGb 761 exerts significant anti-neuroinflammatory effects characterized by a reduction in expression and protein release of cytokines and chemokines in LPS-stimulated BV2 microglial cells, which might be mediated by inhibition of PKC/MAPK and NF-κB signaling. These findings highlight the therapeutic potential of EGb 761 in diseases characterized by neuroinflammation, including AD and other brain injuries and degenerative conditions.

## 6. Outlook

We suggest to further analyze the effect of *Ginkgo biloba* extract EGb 761 on ROS pathways (Nrf2, SOD1/2, etc.) and important parameters of aging such as AMPK and mTOR. Furthermore, we suggest to study effects of EGb 761 on viral-induced neuroinflammation in neuronal and microglial cells. Another topic of interest might be to elucidate if the anti-neuroinflammatory effects are mediated by the proanthocyanidins in the extract.

## Figures and Tables

**Figure 1 ijms-25-08108-f001:**
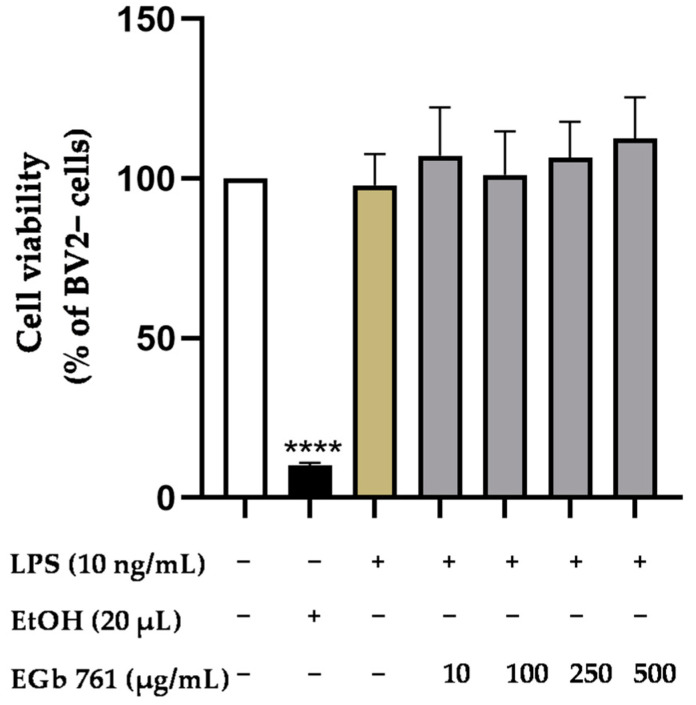
Effects of EGb 761 on cell viability in LPS-stimulated BV2 microglial cells. Cell viability was measured after 24 h of treatment by color change due to MTT reduction. Values are presented as mean ± SD of four independent experiments. Statistical analysis was performed using one-way ANOVA with Dunnett’s post hoc test with **** *p* < 0.0001 compared to untreated cells.

**Figure 2 ijms-25-08108-f002:**
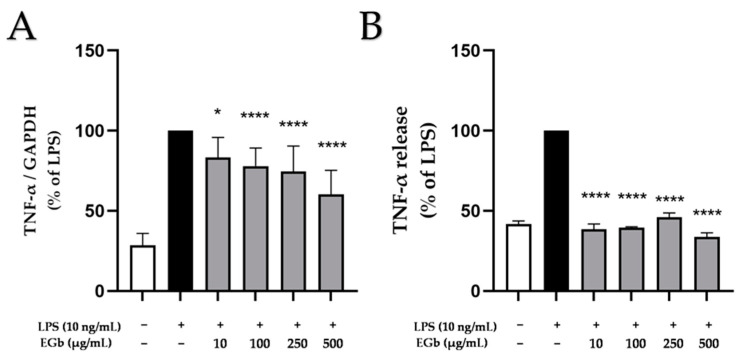
Effects of EGb 761 on TNFα expression (**A**) and release (**B**) in LPS-stimulated BV2 cells. Cells were stimulated as described in Materials and Methods. (**A**) After 4 h of stimulation, RNA was isolated and expression of TNFα was determined using qPCR. (**B**) After 24 h of stimulation, supernatants were collected and the release of TNFα was measured by ELISA. Values are presented as the mean ± SD of nine and three independent experiments for qPCR and ELISA, respectively. Statistical analysis was performed using one-way ANOVA with Dunnett’s post hoc tests with * *p* < 0.05 and **** *p* < 0.0001 compared to LPS.

**Figure 3 ijms-25-08108-f003:**
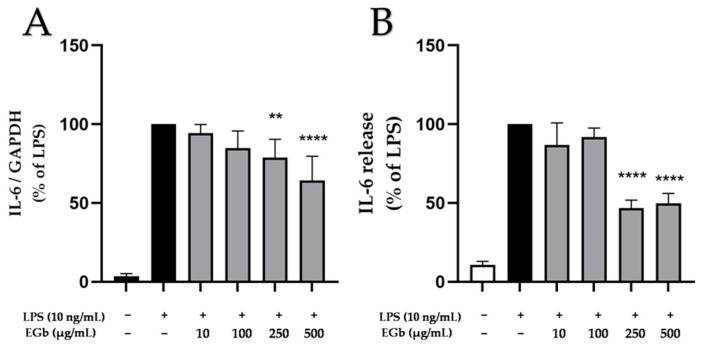
Effects of EGb 761 on IL-6 expression (**A**) and synthesis (**B**) in LPS-stimulated BV2 cells. Cells were stimulated as described in Materials and Methods. (**A**) After 4 h of stimulation, RNA was isolated and expression of IL-6 was determined using qPCR. (**B**) After 24 h of stimulation, supernatants were collected and the release of IL-6 was measured by ELISA. Values are presented as the mean ± SD of five and three independent experiments for qPCR and ELISA, respectively. Statistical analysis was performed using one-way ANOVA with Dunnett’s post hoc tests with ** *p* < 0.01, and **** *p* < 0.0001 compared to LPS.

**Figure 4 ijms-25-08108-f004:**
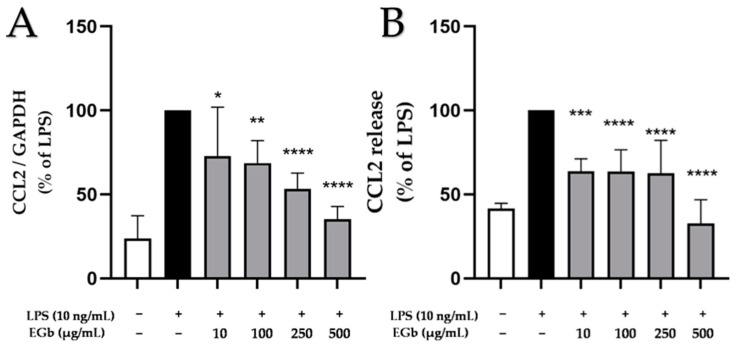
Effects of EGb 761 on CCL2 expression (**A**) and synthesis (**B**) in LPS-stimulated BV2 cells. Cells were stimulated as described in Materials and Methods. (**A**) After 4 h of stimulation, RNA was isolated and expression of CCL2 was determined using qPCR. (**B**) After 24 h of stimulation, supernatants were collected and the release of CCL2 was measured by ELISA. Values are presented as the mean ± SD of six independent experiments. Statistical analysis was performed using one-way ANOVA with Dunnett’s post hoc tests with * *p* < 0.05, ** *p* < 0.01, *** *p* < 0.001 and **** *p* < 0.0001 compared to LPS.

**Figure 5 ijms-25-08108-f005:**
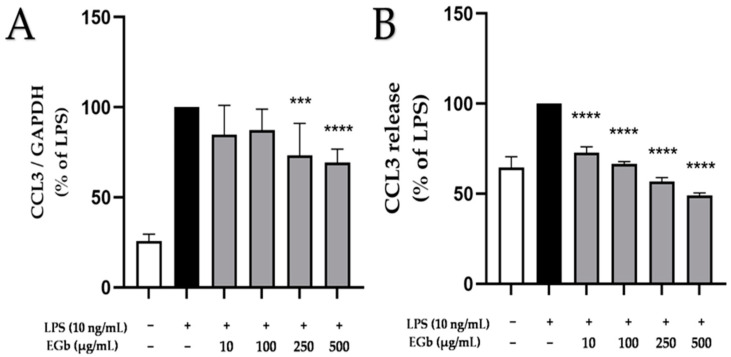
Effects of EGb 761 on CCL3 expression (**A**) and synthesis (**B**) in LPS-stimulated BV2 cells. Cells were stimulated as described in Materials and Methods. (**A**) After 4 h of stimulation, RNA was isolated and expression of CCL3 was determined using qPCR. (**B**) After 24 h of stimulation, supernatants were collected and the release of CCL3 was measured by ELISA. Values are presented as the mean ± SD of nine and three independent experiments for qPCR and ELISA, respectively. Statistical analysis was performed using one-way ANOVA with Dunnett’s post hoc tests with *** *p* < 0.001 and **** *p* < 0.0001 compared to LPS.

**Figure 6 ijms-25-08108-f006:**
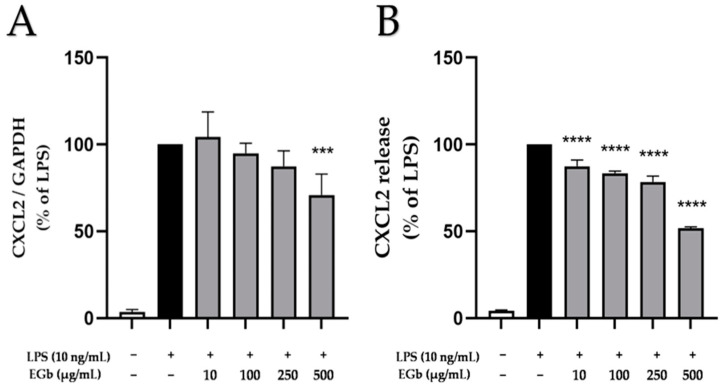
Effects of EGb 761 on CXCL2 expression (**A**) and protein release (**B**) in LPS-stimulated BV2 cells. Cells were stimulated as described in Materials and Methods. (**A**) After 4 h of stimulation, RNA was isolated and expression of CXCL2 was determined using qPCR. (**B**) After 24 h of stimulation, supernatants were collected and the release of CXCL2 was measured by ELISA. Values are presented as the mean ± SD of six independent experiments. Statistical analysis was performed using one-way ANOVA with Dunnett’s post hoc tests with *** *p* < 0.001 and **** *p* < 0.0001 compared to LPS.

**Figure 7 ijms-25-08108-f007:**
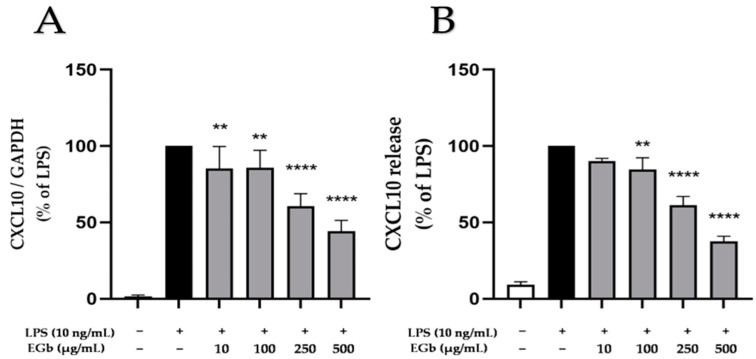
Effects of EGb 761 on CXCL10 expression (**A**) and release (**B**) of CXCL10 in LPS-stimulated BV2 cells. Cells were stimulated as described in Materials and Methods. (**A**) After 4 h of stimulation, RNA was isolated and expression of CXCL10 was determined using qPCR. (**B**) After 24 h of stimulation, supernatants were collected and the release of CXCL10 was measured by ELISA. Values are presented as the mean ± SD of nine and three independent experiments for qPCR and ELISA. Statistical analysis was performed using one-way ANOVA with Dunnett’s post hoc tests with ** *p* < 0.01 and **** *p* < 0.0001 compared to LPS.

**Figure 8 ijms-25-08108-f008:**
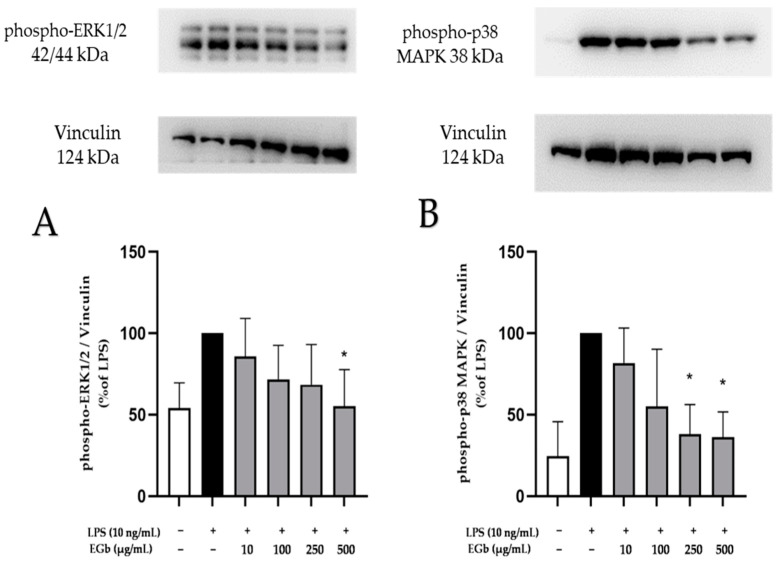
Effects of EGb 761 on the phosphorylation of ERK 1/2 (**A**) and p38 MAPK (**B**) in LPS-stimulated BV2 cells. Cells were stimulated and Western blot was performed as described in Materials and Methods. Values are presented as the mean ± SD of three independent experiments, and protein levels were referenced to Vinculin. Statistical analysis was performed using one-way ANOVA with Dunnett’s post hoc tests with * *p* < 0.05 compared to LPS.

**Figure 9 ijms-25-08108-f009:**
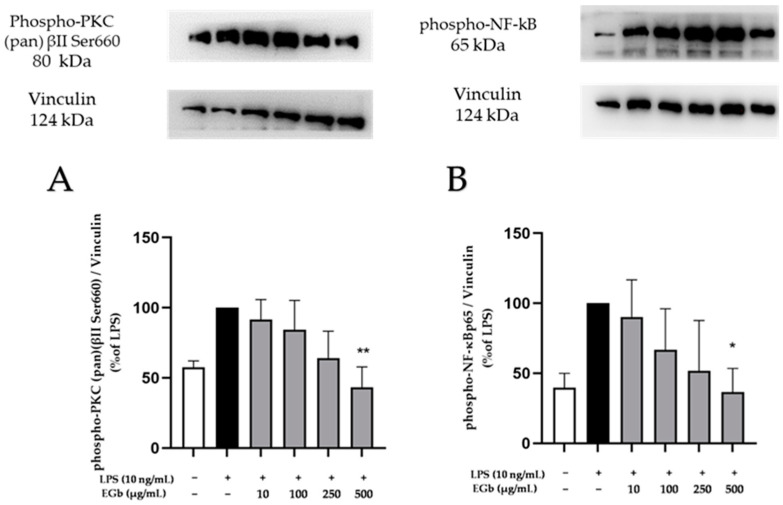
Effects of EGb 761 on phosphorylation of PKC (pan) (βII Ser660) (**A**) and NF-κBp65 (**B**) in LPS-stimulated BV2 cells. Cells were stimulated and Western blot was performed as described in Materials and Methods. Values are presented as the mean ± SD of three independent experiments, and protein levels were referenced to Vinculin. Statistical analysis was performed using one-way ANOVA with Dunnett’s post hoc tests with * *p* < 0.05 and ** *p* < 0.01 compared to LPS.

## Data Availability

The data presented in this manuscript are available from the corresponding author upon request.

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
