# Peer review of "Anti-Neuroinflammatory Effects of Ginkgo biloba Extract EGb 761 in LPS-Activated BV2 Microglial Cells"

_ijms, 2024, doi:10.3390/ijms25158108_

Round 1
Reviewer 1 Report
Comments and Suggestions for Authors
The manuscript entitled “Anti-Neuroinflammatory Effects Of Ginkgo biloba Extract EGb 2
761 in LPS-Activated BV2 Microglial Cells” by Lu Sun et al., was well written and presented. The authors hypothesized Anti-Neuroinflammatory Effects Of Ginkgo biloba Extract EGb 2 761 in LPS-Activated BV2 Microglial Cells by evaluating various inflammatory markers and they succeeded in that.
However, few queries authors need to answer
1. The results presented here were clearly discussed and cited with references.
2. Since the authors mentioned their previous publication, “Anti-neuroinflammatory effects of Ginkgo biloba extract EGb761 in LPS-activated primary microglial cells” , How far the present paper differs from earlier one ?
3. Please change the tile , since both manuscripts title looks similar
4. Is there any mechanism, how the extract crosses blood-brain barrier
5. Is there any half life-PK, PD datas?
6. iNOS might playing important role, if possible discuss two lines about that
7. If possible authors may include schematic figure, as like previous publication
8. The immunoblots results are good- How many replicates performed for each protein
9. How far this work may translatable to pre clinical studies
10. What is the rational or choosing Ginkgo biloba Extract EGb 2 761
Author Response
Reviewer 1
The manuscript entitled “Anti-Neuroinflammatory effects of Ginkgo biloba extract EGb761 in LPS-activated BV2 microglial cells” by Lu Sun et al., was well written and presented. The authors hypothesized Anti-Neuroinflammatory Effects of Ginkgo biloba Extract EGb 761 in LPS-Activated BV2 Microglial Cells by evaluating various inflammatory markers and they succeeded in that.
However, few questions authors need to answer
Q1. The results presented here were clearly discussed and cited with references.
Response: Thank you for your comment
Q2. Since the authors mentioned their previous publication, “Anti-neuroinflammatory effects of Ginkgo biloba extract EGb761 in LPS-activated primary microglial cells” , How far the present paper differs from earlier one ?
Response: In our previous study, we used primary microglial cells and were able to show an inhibitory effect of Ginkgo biloba extract EGb761 on the COX-2/PGE2 pathway. In the current study, we focused on effects of EGb 761 on different cytokines and chemokines, providing additional information on the anti-inflammatory effects of the investigated extract. We decided to use BV-2 microglial cells in the current study to avoid suffering of animals, since BV-2 microglial cells are known to be a well established model for microglial research [1].
Q3. Please change the title, since both manuscripts title looks similar
Response: We believe, that the title provides the necessary information. Surely, it is quite similar to the previous published study, however, the different cells used are represented in the title. Both studies present anti-neuroinflammatory effects, only the pathways/targets targeted in the studies vary.
Q4. Is there any mechanism, how the extract crosses blood-brain barrier
Response: EGb 761 contains active compounds such as flavonoids and terpenoids. There components are lipophilic and small enough to cross the BBB. Moreover, it is published that EGb 761 may enhance BBB permeability to its components [2].
Q5. Is there any half life-PK, PD datas?
Response: The plant extract EGb 761 contains about 300 herbal ingredients. It is not known for each individual substance how long they remain in the body. Moreover, for drugs with CNS effects, bioavailability at the site of action is difficult to demonstrate. Nevertheless, this has been achieved for several ingredients of EGb 761. Orally administered EGb 761 does show a high bioavailability for A and B ginkgolides and bilobalide (80-90%) and its serum peak concentration is reached after 1-2 hours. The plasma half-life times vary from 4 h (ginkgolide A and bilobalide) to 10 h (ginkgolide B). The half-life of the flavonoids is between 10-17 h. The compounds of EGb 761 are primarily eliminated via urine. An interaction with isoenzymes of the cytochrome P450 family has also been shown [3]. In humans, the cerebral bioavailability of EGb 761 has been demonstrated by a dose-dependent influence on brain electrical activity in the pharmaco-EEG [4–6].
Q6. iNOS might playing important role, if possible discuss two lines about that
Response: Thank you for your comment. We included a short section to observed effects of EGB 761 on iNOS (lines 686-688).
Q7. If possible authors may include schematic figure, as like previous publication
Response: Thank you for your suggestion. We added the graphical abstract in the manuscript, since reviewer do not have access to this graphic during the review process.
Q8. The immunoblots results are good- How many replicates performed for each protein
Response: All immunoblot results present three independent experiments for all investigated kinases. The number of replicates is provided in the figure legends.
Q9. How far this work may translatable to pre clinical studies
Response: For the current study, we used BV2 microglial cells to exert anti-inflammatory effects of EGb 761. Due to the positive results, further research in complex cell-based models, such as mixed glial cultures or organotypical hippocampal slice cultures (OHSC), might offer new insights on the effects of EGb 761 with respect to cell-interactions. Furthermore, effects of EGb 761 on neuroinflammatory processes could additionally be examined in wildtype murine models as well as specific inflammatory disease models to evaluate effects on disorder-specific symptoms. A possible transition requires careful consideration of dosage, bioavailability, and the biological relevance of the models used.
Q10. What is the rational or choosing Ginkgo biloba Extract EGb 761
Response: We chose Ginkgo extract EGb 761 in our experiments because it is among the best characterized plant extracts in the world [7]. Moreover, it is the only Ginkgo extract specifically recommended in international clinical guidelines [8].
- Henn, A. The Suitability of BV2 Cells as Alternative Model System for Primary Microglia Cultures or for Animal Experiments Examining Brain Inflammation. ALTEX 2009, 83–94, doi:10.14573/altex.2009.2.83.
- Liang, W.; Xu, W.; Zhu, J.; Zhu, Y.; Gu, Q.; Li, Y.; Guo, C.; Huang, Y.; Yu, J.; Wang, W.; et al. Ginkgo Biloba Extract Improves Brain Uptake of Ginsenosides by Increasing Blood-Brain Barrier Permeability via Activating A1 Adenosine Receptor Signaling Pathway. Journal of Ethnopharmacology 2020, 246, 112243, doi:10.1016/j.jep.2019.112243.
- McKeage, K.; Lyseng-Williamson, K.A. Ginkgo Biloba Extract EGb 761® in the Symptomatic Treatment of Mild-to-Moderate Dementia: A Profile of Its Use. Drugs Ther Perspect 2018, 34, 358–366, doi:10.1007/s40267-018-0537-8.
- Itil, T.M.; Eralp, E.; Tsambis, E.; Itil, K.Z.; Stein, U. CENTRAL NERVOUS SYSTEM EFFECTS OF GINKGO BILOBA, A PLANT EXTRACT: American Journal of Therapeutics 1996, 3, 63–73, doi:10.1097/00045391-199601000-00009.
- Itil, T.; Martorano, D. Natural Substances in Psychiatry (Ginkgo Biloba in Dementia). Psychopharmacol Bull 1995, 31, 147–158.
- Maurer, K.; Ihl, R.; Dierks, T.; Frölich, L. Clinical Efficacy of Ginkgo Biloba Special Extract EGb 761 in Dementia of the Alzheimer Type. Journal of Psychiatric Research 1997, 31, 645–655, doi:10.1016/S0022-3956(97)00022-8.
- Kulić, Ž.; Wolff, J.; Wilhelm, E.; Schüler, V.; Röck, B.; Butterer, A. Sourcing Shikimic Acid from Waste Streams of Ginkgo Biloba Leaf Extract Production. ACS Sustainable Chem. Eng. 2023, 11, 4943–4947, doi:10.1021/acssuschemeng.3c00585.
- Kasper, S.; Bancher, C.; Eckert, A.; Förstl, H.; Frölich, L.; Hort, J.; Korczyn, A.D.; Kressig, R.W.; Levin, O.; Palomo, M.S.M. Management of Mild Cognitive Impairment (MCI): The Need for National and International Guidelines. The World Journal of Biological Psychiatry 2020, 21, 579–594, doi:10.1080/15622975.2019.1696473.
Reviewer 2 Report
Comments and Suggestions for Authors
In this study, the authors explored the potential anti-neuroinflammatory effect of an extract from Ginkgo biloba leaves in BV2 microglial cells that were stimulated with LPS. The authors demonstrated that this effect was primarily achieved through the inhibition of several pro-inflammatory cytokines. The results are significant and of interest to the audience. However, there are some points that the authors should address.
- The abstract should include the specific concentrations of the Ginkgo biloba extract and LPS used, as well as the treatment periods.
- Some keywords could be altered to differ from those in the title.
- In lines 37-38, depression is cited alongside Alzheimer’s and Parkinson’s disease as a neurodegenerative disease, rather than a psychiatric disorder. Please correct.
- Using only the MTT assay to assess cell viability is a methodological constraint of this study. This assay is known to have bias, so another test should be conducted to confirm these results. Additionally, I am confused as to why the authors only included two concentrations of the extract (250 and 500 μg/ml) in Figure 1, while later results show four concentrations (10 to 500 μg/ml). It would be more informative to include this broader range in Figure 1 as well. Please review these issues carefully.
- The LPS group in the figures is missing error bars, and Figure 1 lacks symbols to indicate significant differences.
- There is a need to further explain why the authors chose these specific concentrations of the extract and exposure periods.
- I believe there are too many figures with only one or two graphs each. The figures could be reorganized, grouping some of them, to make the text more attractive. Perhaps including a graphical abstract or representative figure of the mechanisms of action of the extract would be helpful for future readers.
- Vinculin is a rather unusual choice of protein control for western blot. Why this?
- “Ginkgo biloba” should be written in italics throughout the text.
Author Response
Reviewer 2
In this study, the authors explored the potential anti-neuroinflammatory effect of an extract from Ginkgo biloba leaves in BV2 microglial cells that were stimulated with LPS. The authors demonstrated that this effect was primarily achieved through the inhibition of several pro-inflammatory cytokines. The results are significant and of interest to the audience. However, there are some points that the authors should address.
- The abstract should include the specific concentrations of the Ginkgo biloba extract and LPS used, as well as the treatment periods.
Response: We appreciate your comment and added the concentrations of EGb 761 in the abstract accordingly (line 26). However, we believe that the specific treatment periods, that differ for immunoblots, ELISAs and qPCRs might lead to confusion when mentioned in the abstract and would suggest to refer to the Material and method or results section for that information.
- Some keywords could be altered to differ from those in the title
Response: Thank you for your comment, we altered the keywords.
- In lines 37-38, depression is cited alongside Alzheimer’s and Parkinson’s disease as a neurodegenerative disease, rather than a psychiatric disorder. Please correct.
Response: Thank you for your comment, we changed the term neurodegenerative to neuropsychiatric, that summarize the mentioned diseases in a much better way.
- 4. Using only the MTT assay to assess cell viability is a methodological constraint of this study. This assay is known to have bias, so another test should be conducted to confirm these results. Additionally, I am confused as to why the authors only included two concentrations of the extract (250 and 500 μg/ml) in Figure 1, while later results show four concentrations (10 to 500 μg/ml). It would be more informative to include this broader range in Figure 1 as well. Please review these issues carefully
Response: The MTT tetrazolium assay technology has become the gold standard for the determination of cell viability and proliferation since its development by Mosmann in the 1980s compared to another assay. MTT is not only the gold standard for cytotoxicity testing but also suits for high-throughput screening and miniaturization [9,10]. Therefore, MTT assay are reliable and valid. However, MTT assay conversion to formazan crystals depends on metabolic rate and number of mitochondria resulting in many known interferences [11]. The authors know about the influences of cell metabolism on the results of MTT assay due to reduction and oxidation processes. Since the assay was performed on a 96-well plate in the same cell passage and further conditions were the same for all cells, we do not expect reasons for altered cell metabolism due to environmental factors.
Furthermore, we investigated possible cytotoxic effects of all used concentrations of EGb 761 in the MTT assay. We added the two lower concentrations 10 and 100 µg/mL in the revised version of the manuscript.
- The LPS group in the figures is missing error bars, and Figure 1 lacks symbols to indicate significant differences.
Response: The LPS group does not display error bars due to our way to calculate the statistics. We set the respective control in each independent experiment to 100% and calculate the other experimental groups in respect to the control group. The significance of the ethanol group was added to figure 1.
- There is a need to further explain why the authors chose these specific concentrations of the extract and exposure periods.
Response: We orientated the concentrations used of the current study on our previous study in primary microglial cells. The tested compounds did not exert any cytotoxic effects in this doses range. The normal oral dosage of EGb 761 in humans is 120-240 mg/d. Due to its high bioavailability of 80-90%, we orientated on these concentrations [3], too, and added another higher concentration to evaluate any additional effects of higher dosage in vitro. The specific exposure times are derived of our laboratory experience in the investigation of different chemokines and cytokines on expression and release levels as well as the investigation of the MAPK pathway in immunoblots.
- I believe there are too many figures with only one or two graphs each. The figures could be reorganized, grouping some of them, to make the text more attractive. Perhaps including a graphical abstract or representative figure of the mechanisms of action of the extract would be helpful for future readers.
Response: Thank you for your comment. We believe that presenting the figures separately offers a better overview and allows easy comparison of the effects of EGb 761 on release and expression (ELISA/qPCR). A graphical abstract was submitted with the paper, but it is not available to the reviewers. We now added the graphical abstract to the manuscript.
- Vinculin is a rather unusual choice of protein control for western blot. Why this?
Response: Vinculin is a constitutively expressed protein that maintains relatively stable levels across different experimental conditions and cell types. Vinculin levels may remain relatively stable upon activation of NF-kappaB or MAPK pathways [12]. Therefore, its expression is less likely to be influenced by signalling-induced changes compared to tubulin, GAPDH, or actin. In addition, vinculin is abundantly expressed and validated in the literature. Using vinculin as a reference protein ensures normalization to a control that reflects the signaling environment in these subcellular compartments.
- “Ginkgo biloba” should be written in italics throughout the text.
Response: We appreciate your comment and revised the text accordingly.
References
9. Van Tonder, A.; Joubert, A.M.; Cromarty, A.D. Limitations of the 3-(4,5-Dimethylthiazol-2-Yl)-2,5-Diphenyl-2H-Tetrazolium Bromide (MTT) Assay When Compared to Three Commonly Used Cell Enumeration Assays. BMC Res Notes 2015, 8, 47, doi:10.1186/s13104-015-1000-8.
10. Hamid, R.; Rotshteyn, Y.; Rabadi, L.; Parikh, R.; Bullock, P. Comparison of Alamar Blue and MTT Assays for High Through-Put Screening. Toxicology in Vitro 2004, 18, 703–710, doi:10.1016/j.tiv.2004.03.012.
11. Lü, L.; Zhang, L.; Wai, M.S.M.; Yew, D.T.W.; Xu, J. Exocytosis of MTT Formazan Could Exacerbate Cell Injury. Toxicology in Vitro 2012, 26, 636–644, doi:10.1016/j.tiv.2012.02.006.
12. Carisey, A.; Ballestrem, C. Vinculin, an Adapter Protein in Control of Cell Adhesion Signalling. European Journal of Cell Biology 2011, 90, 157–163, doi:10.1016/j.ejcb.2010.06.007.